# Equivalent user experience and improved community augmented meta-analyses knowledge for a new version of a Plain Language Summary guideline

**Mark Jonas** ⓘ *, **Martin Kerwer, Marlene Stoll, Gesa Benz, Anita Chasiotis**

Leibniz Institute for Psychology (ZPID), Trier, Germany

* mj@leibniz-psychology.org

**Data Availability Statement:** All relevant data are available in the paper and its Supporting Information files. Additionally, the dataset, codebook, complete R markdown, and shortened

## Abstract

Plain Language Summaries (PLS) offer a promising solution to make meta-analytic psychological research more accessible for non-experts and laypeople. However, existing writing guidelines for this type of publication are seldom grounded in empirical studies. To address this and to test two versions of a new PLS guideline, we investigated the impact of PLSs of psychological meta-analyses on laypeoples' PLS-related knowledge and their user experience (accessibility, understanding, empowerment). In a preregistered online-study, $N = 2,041$ German-speaking participants read two PLSs. We varied the inclusion of a disclaimer on PLS authorship, a statement on the causality of effects, additional information on community augmented meta-analyses (CAMA) and the PLS guideline version. Results partially confirmed our preregistered hypotheses: Participants answered knowledge items on CAMA more correctly when a PLS contained additional information on CAMA, and there were no user experience differences between the old and the new guideline versions. Unexpectedly, a priori hypotheses regarding improved knowledge via the use of a disclaimer and a causality statement were not confirmed. Reasons for this, as well as general aspects related to science communication via PLSs aimed at educating laypeople, are discussed.

## Introduction

Communicating the main points of a study or research article to individuals without a scientific background is often challenging. Research articles frequently include methodological details or scientific jargon unfamiliar to laypeople [1]. Plain Language Summaries (PLSs) offer one possible solution to this issue: As comprehensible, lay-friendly summaries of research, they aim to make research findings accessible for the general public and to improve scientific understanding. So far, PLSs have mostly been applied in the medical field [2], but may also have potential for other fields, such as psychology. However, empirically validated guidelines for writing PLSs of psychological research are lacking and guidance on writing PLSs is based on empirical studies in very few cases only [3]. To our knowledge, this lack of empirical studies

R markdown are available at the following DOIs: https://doi.org/10.23668/psycharchives.14218, https://doi.org/10.23668/psycharchives.14219.

**Funding:** The study was funded by internal ZPID funds. All authors were employed at the ZPID during the runtime of the study. The authors received no additional grants or specific funding for this work.

**Competing interests:** The authors have declared that no competing interests exist.

particularly applies to PLSs on psychological meta-analyses. Yet, an empirical base for communicating the aggregated, higher level evidence of meta-analyses seems especially crucial. To address this issue, a new writing guideline for psychological PLSs (the KLARpsy guideline [4]) has been developed at the Leibniz Institute for Psychology (ZPID) and published in an open access repository. In the context of project"PLan Psy", several experimental studies were conducted to validate older versions of the guideline [5, 6]. The objective is to make PLSs of psychological meta-analyses based on this guideline freely available to the German public as part of the information service "KLARpsy" (hence the guideline name; "Klar" is the German word for "clear" or "plain"). To that end, PLSs will be made available at the website klarpsy.de, starting in October 2023. The guideline may also serve other researchers interested in communicating scientific psychology to the public. Guideline development was subject to an iterative improvement process, during which user feedback from qualitative studies [7] and from an expert survey [8] led us to add new elements to the initial version of the guideline. These elements included more detailed information on the authorship of the PLSs and meta-analyses, a statement on the causality of effects and information on community-augmented meta-analyses (CAMA). However, since our intention is to write PLSs according to a guideline with a strong empirical basis, our aim was to compare both guidelines in a large experimental study with regard to PLSs user-relevant outcomes. Furthermore, since empirical research on PLS guidelines or writing criteria is still rare, this may be even more true for studies systematically comparing individual PLS versions based on different guidelines or guideline versions [3]. Even if a PLSs guideline has already been developed to an advanced stage, further adjustments should ideally be empirically tested in a large target group sample. Otherwise, it is not possible to rule out that changes are merely implemented based on idiosyncratic opinions and do not take reader interests and needs into account.

The study presented here therefore specifically aims to contrast an older version of the guideline with a new and revised version based on user feedback and an expert survey. Following preregistration (available via PsychArchives under the following link: https://doi.org/10.23668/psycharchives.8251), it was conducted online and systematically varied 1) whether a *disclaimer* on the extent of evaluation provided by the PLSs' authors was included in the PLS, 2) whether a *causality statement* containing additional information on the causal interpretation of effects was included in the PLS, 3) whether *additional information on Community Augmented Meta-Analyses* (CAMA) were available and 4) whether PLSs were created based on a *new or old guideline version.*

Outcomes include several knowledge items, e.g. on the relationship between PLS and meta-analyses or on the extent of evaluation carried out by PLS authors regarding the original study. Furthermore, readers' user experience (accessibility, understanding, empowerment), their perceived epistemic trustworthiness of meta-analyses authors as well as PLS authors, their perceptions regarding the credibility of the presented evidence and the perceived personal relevance of research findings were examined.

So far, empirical evidence on whether laypeople can distinctly differentiate between the two formats of an original meta-analysis and a subsequent PLS is scarce. Ideally, laypeople should be able to grasp that a PLS summarizes findings of a more extensive research work in an accessible manner for an audience not exclusively rooted in science. Knowledge of this fact is relevant for multiple reasons: First, PLSs may be written by third-party authors not involved in the original studies. These third-party authors may emphasize different aspects of a main study's results in their PLS, e.g. by highlighting them more prominently or mentioning them first. For example, authors may be more inclined to spotlight treatment outcomes for one particular approach of psychotherapy compared to other approaches, depending on their work approach or therapeutic background. And second, PLSs often aim to concisely convey the key points of

a study. However, this may come with the potential risk of leaving out more specific information (e.g. the effectiveness of a therapy approach for a specific subgroup or higher-level interaction effects). Both issues may nudge readers' process of informed decision-making in a certain direction. With this in mind, is there a way of supporting laypeople in grasping author differences between an original study and its subsequent PLS? One possible solution could be to explicitly state the difference between the two formats via a disclaimer. This disclaimer could outline the purpose of a PLS and its relation to the original study, the extent to which a PLS evaluates the quality and rigor of the original study (see H2) and provide details on PLS authors (see H3). We assume that including such a disclaimer in a PLS will support laypeople in comprehending how the PLS and meta-analysis relate to each other, and thus expect the following effects:

*H1a*: If a PLS based on the new guideline contains a disclaimer, readers will score significantly higher on a knowledge item concerning the relationship between PLS and meta-analysis compared to a PLS based on the old guideline.

*H1b*: If a PLS based on the new guideline contains a disclaimer, readers will score significantly higher on a knowledge item concerning the relationship between PLS and meta-analysis compared to a PLS based on the new guideline without a disclaimer.

By default, laypeople could also assume that PLSs are only written for high-quality studies or that the content of the original meta-analyses was specifically evaluated or replicated by the PLS authors or organizations offering PLSs. This "Trust Heuristic" [9] may enable readers to quickly make trust decisions. Yet, it may also lead to misconceptions about the extent of evaluation carried out in the context of the PLS. For example, PLSs in the project "PLan Psy" typically did not evaluate the rigor or correctness of meta-analytic procedures in detail, and also did not carry out replications. In a worst case scenario, laypeople may however assume that such an evaluation took place, which could result in an incorrect assessment of the generalizability of meta-analytic findings presented. Ultimately, this may again influence them in their informed decision-making. Providing laypeople with the above-mentioned disclaimer may help them to more accurately assess the conclusions that can and cannot be drawn from PLSs. Following this, we propose:

*H2a*: If a PLS based on the new guideline contains a disclaimer, readers will score significantly higher on a knowledge item concerning the extent of evaluation compared to a PLS based on the old guideline.

*H2b*: If a PLS based on the new guideline contains a disclaimer, readers will score significantly higher on a knowledge item concerning the extent of evaluation compared to a PLS based on the new guideline without a disclaimer.

Previous research demonstrated that laypeople often vary with regard to how much attention they pay to sources of scientific claims to evaluate information [10]. For instance, Barzilai et al. [11] summarize multiple previous studies and point out that laypeople can distinguish between different types of sources, but that especially novices rarely consider source features. When applying these findings to the format of PLSs, the following question emerges: Are laypeople able to grasp that PLS can stem from an additional set of independent authors who did not conduct the original meta-analysis [12]? Given the fact that source features are rarely considered, this may be challenging to comprehend. Readers may simply assume that both texts stem from the researchers of the original meta-analysis. However, this assumption may pose risks, such as failing to grasp that a different set of authors may influence the level of priority given to particular results. A solution for this could be to once again draw on the above-mentioned disclaimer to make authorship differences salient. We assume that including this information will help to highlight differences in authorship for lay readers and thus expect the following:

*H3a*: If a PLS based on the new guideline contains a disclaimer, readers will score significantly higher on a knowledge item concerning the differentiation between PLS and meta-analysis authors compared to a PLS based on the old guideline.

*H3b*: If a PLS based on the new guideline contains a disclaimer, readers will score significantly higher on a knowledge item concerning the differentiation between PLS and meta-analysis authors compared to a PLS based on the new guideline.

When confronted with research summaries, it may pose a challenge for laypeople to differentiate between correlational (e.g., studied in cross-sectional studies) and causal (e.g., studied in randomized controlled experiments) relationships between two variables. Not differentiating between these relations can be problematic, as it can give rise to logical fallacies such as "cum hoc ergo propter hoc" [13] and render readers prone to draw incorrect conclusions (e.g., assuming that more sports automatically leads to higher well-being when the effects, in reality, can also be explained by the fact that people who feel better will do more sports than people who feel worse). Laypeople may potentially benefit from additional explanations on the causality of effects in the context of PLSs, and may thus be able to more correctly grasp the reported association between two variables. As such, we propose:

*H4a*: If a PLS based on the new guideline contains additional information on causality, readers will score significantly higher on a knowledge item concerning the causality of effects compared to a PLS based on the old guideline.

*H4b*: If a PLS based on the new guideline contains additional information on causality, readers will score significantly higher on a knowledge item concerning the causality of effects compared to a PLS based on the new guideline.

Community-Augmented Meta-Analyses (CAMA) [14] refer to an approach where a meta-analysis' data-base is stored in an open-access repository. As such, the base of the analysis is dynamic and can be expanded on by community input even after the original analysis has been carried out. In addition, the analysis can quickly be replicated via a graphical user-interface and online analysis tools. In comparison to a traditional meta-analysis, this allows for a more dynamic approach to replication and helps to create "living evidence" [15]. This may offer benefits for laypeoples' decision-making process. Normally, a meta-analysis' sample remains set after publication, cannot be dynamically updated and may therefore no longer represent the current state of research, as new study results become available. Relying on CAMA benefits laypeople in that they are more likely to receive up-to-date results and in that they can check effects for their consistency. However, laypeople are likely unaware of the particular features of CAMA in comparison to a more traditional form of meta-analyses. As such, it seems worthwhile to examine how their awareness and knowledge of this particular form of meta-analysis can be increased. Providing them with additional information on CAMA in the context of a PLS should have a positive impact on their knowledge gain. We therefore hypothesize the following:

*H5a*: If a PLS based on the new guideline contains additional information on CAMA, readers will score significantly higher on a knowledge item concerning CAMA compared to a regular PLS based on the old guideline.

*H5b*: If a PLS based on the new guideline contains additional information on CAMA, readers will score significantly higher on a knowledge item concerning CAMA compared to a regular PLS based on the new guideline.

As mentioned above, a new KLARpsy guideline version was created based on an old KLARpsy guideline version. The old version constitutes a pilot version [16] and was compiled based on the results of a systematic literature review by Stoll et al. [3] and an initial study by Kerwer, Chasiotis, et al. [17]. This version was subsequently evaluated by experts with a background in science communication, meta-analyses and/or psychological publishing [8] and was

revised based on their feedback to create the new version of the guideline [18]. While the new version contains some additional information (e.g. the disclaimer on the extent of evaluation or information on causality), other aspects of the guideline were streamlined. As such, laypeoples' user experience (i.e., their ratings of accessibility, understanding and empowerment, see Kerwer, Stoll, et al. [5]) while reading PLSs based on the new version of the guideline should not be inferior to the experience of reading PLSs based on the old guideline. We hence assume:

*H6*: Reading a PLS based on the new guideline will yield a significant non-inferiority test on user experience compared to reading a PLS based on the old guideline.

In addition to these confirmatory hypotheses, some exploratory research questions mainly informed by user feedback from our previous studies were also of interest. How research is funded constitutes an important information for laypeople, in that it directly informs their judgements of epistemic trustworthiness and "second hand evaluations" of science [19]. For instance, laypeople tend to show increased vigilance towards scientists if they are funded by private companies in comparison to public institutions [20]. Findings from the science barometer 2022 [21], a representative German population survey, show that 56% of participants agreed to dependency on funders as a reason to mistrust scientists. Both versions of the KLARpsy guideline explicitly stated how the meta-analysis was funded, yet the exact information surrounding the funding statement was slightly changed. As such, we were interested in potential knowledge differences regarding funding between them.

*RQ1*: Will scores on a knowledge item concerning funding differ if readers receive a PLS based on the new guideline compared to a PLS based on the old guideline?

Taking the line of thought regarding funding information one step further, it seems especially crucial to inform laypeople about conflicts of interest (COI) that could affect the quality of evidence and contribute to the distortion of meta-analytic results. Indeed, this is already an established practice in existing guidelines [22]. Evidence suggests that laypeople do take conflict of interest, such as financial motives, into account when presented with scientific claims by increasing their epistemic vigilance and adjusting their assessments of epistemic trustworthiness (see Gierth & Bromme [23]). Both guideline versions included statements on COI, yet once again the exact position and wording was adjusted between guidelines. We hence aimed to investigate if these changes had any influence on laypeoples' knowledge regarding COI.

*RQ2*: Will scores on a knowledge item concerning conflict of interest differ if readers receive a PLS based on the new guideline compared to a PLS based on the old guideline?

Epistemic Trustworthiness can be defined as a special form of trust centered around knowledge and knowledge gain [24]. Existing research on trustworthiness measures such as the Muenster Epistemic Trustworthiness Inventory [25] typically distinguishes three aspects of epistemic trustworthiness, namely expertise (the perceived ability/competence of a source in a particular field), integrity (a source's adherence to scientific standards and professional rules of conduct) and benevolence (a source's selflessness and interest in others' well-being). When examining trustworthiness in the context of our two guideline versions, two lines of investigation seem worthwhile. First, it may be interesting to determine if trustworthiness differences emerge between the original meta-analysis authors depending on the guideline version. And second, the same question can also be examined with regard to the PLS authors who edited the original meta-analyses. The two following research questions were thus examined:

*RQ3a*: Will differences in the METI ratings of PLS authors (Expertise, Integrity, Benevolence) emerge if readers receive a PLS based on the new guideline compared to a PLS based on the old guideline?

*RQ3b*: Will differences in the METI ratings of meta-analysis researchers (Expertise, Integrity, Benevolence) emerge if readers receive a PLS based on the new guideline compared to a PLS based on the old guideline?

To summarize, the aim of the present study was to examine the impact of a disclaimer, of a causality statement, of additional information on CAMA and of the version actuality (old vs. new) of the KLARpsy guideline on different knowledge items and user experience. Details regarding materials and methods, sample characteristics and experimental procedure will be provided in the following.

## Materials & methods

Prior to data collection, the study was preregistered at PsychArchives (https://doi.org/10. 23668/psycharchives.8251). All study procedures were approved by the ethics committee of Trier University, Germany.

### Design

To test our hypotheses, we employed a between-subjects design with six conditions (see Table 1). Four factors (i.e., independent variables) were varied in within these conditions: (a) whether a *causality statement* was included or not included, (b) whether a *disclaimer* on the extent of evaluation was included or not included, (c) whether a *CAMA-specific PLS* was presented or not presented, and (d) whether PLSs were based on the old *guideline version* or the new *guideline version*. Since full study materials for PLSs in each experimental conditions are available (see S1 File), we only provide a brief summary of the essential differences between the above-mentioned conditions here: PLSs in "causality statement included"-conditions contained an additional statement detailing the causal or noncausal interpretation of the effects that independent variables in the summarized meta-analysis had on relevant outcomes. PLSs in the "disclaimer included"-conditions contained a disclaimer on the extent of evaluation, outlining that PLS authors only translated the original meta-analysis, did not conduct the meta-analysis themselves, did not rate the meta-analysis in terms of its correctness or the topicality of its results, and did not verify the validity of the knowledge claims put forward in the meta-analysis. In the "CAMA Specific PLS"-conditions, we framed the underlying meta-analysis for one of the two PLSs as CAMA (i.e., we suggested that the reported evidence stems from the PsychOpen CAMA platform, https://cama.psychopen.eu/). Concepts such as "living evidence" or the possibility to continuously update analysis results were also mentioned.

Finally, PLSs in the "old guideline version"-conditions were written based on the initial PLS guideline by Chasiotis et al. [16], while PLSs in the "new guideline version"-condition were written based on an updated version of the guideline [18].

### Sample

A general population sample (*N* = 2,041) was recruited via the panel provider Bilendi & respondi. Participants had to be of legal age, possess German language skills at native speaker level, to have successfully graduated from school, and to have a self-reported interest in psychological research (value of "4" or higher on a 1 to 8 rating scale). Participants currently studying psychology or holding a degree in psychology were excluded from the study. To recruit approximately the same proportion of participants in terms of age group (50.20% 18–44 years, 49.80% 45 years or older, *M* = 45.22, *SD* = 15.23, range = 18–90 years), sex (50.37% women, 49.63% men), and education level (33.56% "Hauptschulabschluss", 33.37% "Mittlere Reife", 33.07% "Hochschulreife"), we applied quotas (see Preregistration for more details). It is noteworthy that this resulted in the collection of a balanced demographic sample, rather than a sample precisely representing the age distributions in the German general population.

 

**Table 1. Descriptive statistics grouped by experimental conditions for all variables used in confirmatory testing.**

| Condition | H Tested | Guideline Version | Causality Statement | Disclaimer | CAMA-specific PLS | Extent of Evaluation | | Differentiation | | Causality | | CAMA | | User Experience | |
|---|---|---|---|---|---|---|---|---|---|---|---|---|---|---|---|
| | | | | | | *n* | *Mdn (Q1,Q3)* | *n* | *Mdn (Q1, Q3)* | *n* | *Mdn (Q1, Q3)* | *n* | *Mdn (Q1,Q3)* | *n* | *M (SD)* |
| 1 | H1b H2b H3b H4b | New | no statement included | no disclaimer included | no CAMA PLS included | 217 | 1 (-1,2) | 216 | 0 (-1,2) | 215 | 0 (-3,2) | - | - | 217 | 5.26 (1.37) |
| 2 | H1a, H1b H2a, H2b H3a, H3b H4b | New | no statement included | disclaimer included | no CAMA PLS included | 249 | 0 (-1,3) | 238 | 0 (-1,2) | 242 | -1 (-3.75, 2) | - | - | 247 | 5.40 (1.45) |
| 3 | H1b H2b H3b H4a, H4b | New | statement included | no disclaimer included | no CAMA PLS included | 213 | 0 (-1, 2) | 216 | 0 (-1,2) | 209 | 0 (-2, 2) | - | - | 207 | 5.36 (1.40) |
| 4 | H1a, H1b H2a, H2b H3a, H3b H4a, H4b H5b | New | statement included | disclaimer included | no CAMA PLS included | 204 | 1 (0, 3.25) | 200 | 0 (-0.25,2) | 196 | 0 (-2, 2) | 193 | 0 (-1, 2) | 204 | 5.27 (1.39) |
| 5 | H5a, H5b | New | statement included | disclaimer included | CAMA PLS ncluded | 232 | 0 (-1, 2) | 234 | 0 (-1,2) | 224 | 0 (-3, 2) | 225 | 1 (-1, 5) | 227 | 5.09 (1.46) |
| 6 | H1a, H2a H3a, H4a H5a | Old | no statement included | no disclaimer included | no CAMA PLS included | 241 | 1 (-1,3) | 239 | 0 (0,2) | 239 | 0 (-2, 2.5) | 239 | 0 (-1, 3) | 239 | 5.37 (1.43) |

Note: H Tested = Hypothesis tests conducted with the respective condition, n = number of participants, Mdn = Median, Q1,Q3 = values for the lower and upper quartile, M = arithmetic mean, SD = standard deviation, Guideline Version: New = PLSs were based on the revised version of the PLanPsy PLS guideline, Old = PLSs were based on the original version of the PLanPsy PLS guideline, Causality Statement: no statement included = no information on the causal interpretation of effects was provided, statement included = an additional statement explains the likely noncausal or causal interpretation of effects, Disclaimer: no disclaimer included = no information about the extent to which PLSs were evaluated is provided, disclaimer included = an additional statement explains that the PLS authors only translated the original study and did not evaluate their quality, CAMA- specific PLS: no CAMA PLS included = both summaries did not include CAMA elements, CAMA PLS included = the PLS based on Färber and Rosendahl was presented as a CAMA-analysis, thereby hinting at concepts such as living evidence and continually updated results.

## Procedure

The study was conducted online using the survey software Unipark. Data collection started on October 21st and ended on November 7th 2022. At the beginning of the experiment, participants provided written informed consent. Each participant was then randomly assigned to one experimental condition and read two PLSs, both structured according to the independent variable specification of the assigned condition (e.g., two PLSs that include a causality statement, no disclaimer and no CAMA-specific elements based on the new guideline version, see Table 1). The PLSs used in this experiment were written in German language by the study authors (see S1 File). The results and effects described in these PLSs were based on results and

 

effects of actual meta-analyses on the topic of resilience (based on Färber and Rosendahl [26]) and on the topic of the efficacy of different psychotherapy interventions for depression treatment (based on Barth et al. [27]). All participants read one PLS per topic during the study. Topic order (i.e., whether participants read a PLS on psychotherapy or on resilience first) was randomized. Participants read each of the two texts for at least 3 minutes and answered the outcome measures on the same webpage. Knowledge items on the relationship between PLS and meta-analysis and on the extent of evaluation were administered once, after reading through the first PLS. Knowledge items concerning the differentiation between authorship of the PLS and meta-analysis authors, funding, conflict of interest, and causality were presented twice, after reading through each individual PLS. Due to a technical error in the Unipark script, data on items for differentiation of authorships for one PLS (i.e., the PLS on resilience by Färber & Rosendahl) had to be discarded. Thus, deviating from our preregistration, data on differentiation items exists only for one PLS. Three knowledge items on CAMA were presented once after the relevant PLS. After each PLS, participants also rated either the trustworthiness of either the meta-analysis or PLS authors (see below) via the Muenster Epistemic Trustworthiness Inventory (METI, [25]). At the end of the experiment, participants completed the awareness check, received a debriefing about the purpose of the study and the experimental variations, and were then redirected to the panel provider.

## Variables

Full information on item texts and response formats as well as information on further exploratory outcomes and covariates is provided in the preregistration of this study.

**Knowledge items.**   As knowledge items, we presented sets of statements regarding the following topics: the relationship between the PLS and the corresponding meta-analysis (relationship knowledge, 8 items), the extent of evaluation (evaluation knowledge, 6 items), the differentiation between PLS authors and meta-analysis authors (differentiation knowledge, 6 items), the funding (knowledge on funding, 6 items), the conflicts of interest (knowledge on COI, 7 items), the causality (causality knowledge, 6 items), and the living evidence in PsychOpen CAMA (CAMA knowledge, 8 items). Participants were asked to indicate whether they deemed the corresponding statement to be true or false for each statement separately."Don't know" was provided as a third response option. For our analyses, we coded correct responses as "1", incorrect responses as "-1" and "don't know" as 0 in a first step. In a second step, we created knowledge item scores as sum scores.

**User experience.**   Similarly to a previous study [5] user experience was assessed based on the dimensions of accessibility, understanding and empowerment by means of 8-point Likert scales. Following the approach from a preregistered paper [28], these three dimensions were merged into a single user experience index. Cronbach's alpha for this index was .83.

**Epistemic trustworthiness (METI): Expertise, benevolence and integrity.**   We utilized the METI [25] to assess participants' trustworthiness judgements. To address both RQ3a and RQ3b, half of our participants were randomly asked to rate the METI dimensions for the meta-analysis authors, and half were asked to rate the PLS authors. In the METI, participants rate the epistemic trustworthiness of the scientists whose work is presented according to 14 adjective pairs on a semantic differential corresponding to the three dimensions expertise (six items), integrity (four items) and benevolence (four items). Cronbach's alpha estimates ranging from .90 to .94 indicate that the internal consistency was very good for all METI scales.

**Credibility of the presented evidence and personal relevance.**   Participants rated the perceived credibility of the presented evidence as well as the perceived personal relevance of the PLS topic on 8-point Likert scales.

**Awareness check.** To ensure that participants read the presented texts and questions in a focused and thorough manner, we used an awareness check based on Gamez-Djokic and Molden [29]. Participants received an introductory text to a short scenario (a famine in a village) and the instruction to leave the following question unanswered to demonstrate awareness. They were then presented with a question and a Likert-scale ranging from 1 to 8. 67.76% of the participants passed this awareness check successfully and did not select any answer option.

## Statistical analysis

**Sample size calculation.** An a priori power analysis was carried out using GPower [30]. Since we aimed to compare the different disclaimer, causality statement and CAMA conditions via Wilcoxon-Mann-Whitney tests, we selected "Means: Wilcoxon-Mann-Whitney test (two groups)" as test family and specified the following parameters: A small effect of $d = .20$, $\alpha = .05$, $\beta = .80$ and an allocation ratio of 1. The analysis indicated that at least $N = 325$ participants per group would be required to achieve a power of $\beta = .80$. Since we planned to test H5a and H5b based on the data of only two experimental conditions (see Table 1), and power was expected to be higher for hypotheses using repeated measurements, we decided to recruit at least 325 participants in each of our six experimental conditions. Thus, we aimed for a total sample size of 1,956 participants.

**Analysis plan.** Data analysis was conducted both for a dataset of all participants who finished the study and for a subset of participants who successfully passed the awareness check. For each hypothesis, we first computed Wilcoxon rank sum tests comparing knowledge item scores for the new guideline including the experimental variations described above (i.e. a disclaimer, causality statement or CAMA information) with the old guideline and the new guideline without these variations. Next, we used the R-package *ordinal* [31] to predict differences in knowledge item scores via manipulation conditions and control variables in logistic regressions. For H4, we report an additional logistic regression analysis with a changed reference group for the causality statement instead of a Wilcoxon test in order to adequately address the repeated measurement of the data. In addition, we investigated differences in participants' user experience ratings between guideline versions with non-inferiority-testing via *equivUMP* [32]. For the non-inferiority-test, user experience ratings from both PLSs will be averaged, and separate analyses for each PLS are available in S4 and S5 Files. Finally, we examined RQ1 and RQ2 via cumulative link mixed-models, and RQ3 via linear regressions.

## Results

Because attentive and thorough processing is especially crucial in the context of comparatively minor textual variations such as the inclusion of a disclaimer or a statement on causality, the following results are based on data from participants who successfully passed the awareness check ($N = 1,383$). Distributions of cases where the awareness check failed across demographic variables such as age, gender or educational background are available in S4 and S5 Files. Analyses for all participants are available in the R Markdown document in S4 File, and a shortened analysis script emphasizing the main results for participants who passed the awareness check is available in S5 File. Table 1 provides an overview of descriptive statistics for all six experimental conditions. In the following, we will outline the analysis results for all confirmatory hypotheses and research questions step by step. We will focus on presenting the results for logistic or mixed model regressions. In the first step, we will report model tests against the null model, with Nagelkerke's $R^2$ [33] as an overall effect size. In a second step, we will then highlight specific predictors within the models. Finally, results of Wilcoxon rank-sum tests may be added to illustrate specific condition differences in a more in-depth manner. The regression

results can be found in Table 2, and Fig 1 depicts medians for each hypothesis and guideline version via boxplots. An overview of all analyses is available in S4 File.

### H1: Guideline version, disclaimer and readers' knowledge on the relationship between PLS and original meta-analysis

An ordinal logistic regression was carried out to examine H1. Readers' overall score on the relationship knowledge item was used as criterion, while two variables specifying disclaimer and guideline versions ("No Disclaimer, New Guideline" & "Disclaimer, New Guideline") were used as predictors. Research Summary Type (Färber and Rosendahl compared to Barth et al.), participants' age, sex, educational background and interest ratings were included as additional control variables. The overall model was significant compared to a null model, $\chi^2$ (11) = 146.68, $p < .001$, $R^2 = 0.124$. No significant influences on the relationship knowledge item scores emerged for PLSs following the new guideline versions, irrespective of the presence or absence of disclaimers (No Disclaimer, New Guideline: OR = 0.94, 95%CI [0.72, 1.23], $p = .663$; Disclaimer, New Guideline: OR = 1.12, 95%CI [0.85, 1.47], $p = .408$). The use of the new version of the guideline as well as the inclusion of a disclaimer did not significantly increase the likelihood of correct answers to the relationship knowledge items compared to the old guideline. In addition, no relationship knowledge score differences in the new guideline condition were found between PLSs including or excluding a disclaimer ($Mdn_{\text{No Disclaimer}} = 0$ vs. $Mdn_{\text{Disclaimer}} = 0$, $W = 90501$, $p = .186$). Therefore, including a disclaimer in the new guideline version neither improved laypeoples' knowledge on the relationship between PLS and original meta-analysis compared to the old guideline (H1a) nor to the regular new guideline (H1b). Thus, H1a and H1b can be rejected.

### H2: Guideline version, disclaimer and readers' knowledge on PLSs extent of evaluation

In order to examine H2, we again employed ordinal logistic regression. Readers' overall scores on the evaluation knowledge item were entered as criterion, and the conditions "No Disclaimer, New Guideline" and "Disclaimer, New Guideline" were once again used as predictors. Furthermore, the above-mentioned control variables were again included in the model. The overall model was significant compared to a null model, $\chi^2$ (11) = 99.78, $p < .001$, $R^2 = .086$. As for the influence of the new guideline version and the presence or absence of a disclaimer, the corresponding predictors were not significant (No Disclaimer, New Guideline: OR = 0.86, 95%CI [0.66, 1.14], $p = .293$; Disclaimer, New Guideline: OR = 1.12, 95%CI [0.85, 1.48], $p = .402$). Participants were not more likely to answer the evaluation knowledge item correctly when they received a PLS based on the new guideline with or without a disclaimer compared to the old guideline. However, a Wilcoxon rank-sum test revealed that the overall evaluation knowledge score was higher in the new guideline condition when a disclaimer was used ($Mdn_{\text{No Disclaimer}} = 0$ vs. $Mdn_{\text{Disclaimer}} = 1$, $W = 89460$, $p < .05$). Including a disclaimer in the new guideline version thus did not improve knowledge on the extent of evaluation compared to the old guideline (H2a), but had positive impacts when compared to the regular version of the new guideline (H2b). H2a is thus rejected, while H2b is confirmed.

### H3: Guideline version, disclaimer and readers' knowledge on the differentiation between PLS authors and MA authors

H3 was examined via ordinal logistic regression, with readers' overall scores on the differentiation knowledge item as a criterion and guideline and disclaimer conditions ("No Disclaimer,

**Table 2. Results of ordinal regression analyses via cumulative link models and cumulative link mixed models for the knowledge items focused on the relationship between the PLS and the corresponding meta-analysis, extent of evaluation, differentiation between PLS authors and meta-analysis authors, funding, conflict of interest, causality, and living evidence in PsychOpen CAMA.**

| Outcome | Model Parameter | EST (SE) | OR | 95% CI |
|---|---|---|---|---|
| **Relationship** | No Disclaimer—New Guideline | - 0.060 (0.138) | 0.941 | 0.718, 1.235 |
| $N_{ID}$ = 1114 | Disclaimer—New Guideline | 0.115 (0.139) | 1.122 | 0.855, 1.472 |
| | Summary–Färber | 0.030 (0.105) | 1.030 | 0.839, 1.266 |
| | Age | - 0.026 (0.004) *** | 0.974 | 0.968, 0.981 |
| | Sex—Male | - 0.197 (0.107) | 0.822 | 0.666, 1.013 |
| | Education—Medium | 0.460 (0.134) *** | 1.584 | 1.219, 2.058 |
| | Education—High | 1.073 (0.134) *** | 2.925 | 2.253, 3.803 |
| | Interest—5 | 0.024 (0.165) | 1.024 | 0.741, 1.415 |
| | Interest—6 | 0.135 (0.166) | 1.145 | 0.827, 1.585 |
| | Interest—7 | - 0.045 (0.179) | 0.956 | 0.673, 1.358 |
| | Interest—8 | - 0.129 (0.178) | 0.878 | 0.620, 1.246 |
| **Extent of Evaluation** | No Disclaimer—New Guideline | - 0.147 (0.140) | 0.863 | 0.655, 1.136 |
| $N_{ID}$ = 1124 | Disclaimer—New Guideline | 0.118 (0.140) | 1.125 | 0.854, 1.482 |
| | Summary—Färber | 0.020 (0.105) | 1.021 | 0.831, 1.254 |
| | Age | - 0.012 (0.003) *** | 0.988 | 0.982, 0.995 |
| | Sex—Male | 0.079 (0.106) | 1.082 | 0.878, 1.333 |
| | Education—Medium | 0.565 (0.135) *** | 1.760 | 1.352, 2.293 |
| | Education—High | 1.102 (0.134) *** | 3.009 | 2.317, 3.915 |
| | Interest—5 | - 0.153 (0.167) | 0.858 | 0.618, 1.190 |
| | Interest—6 | - 0.142 (0.166) | 0.868 | 0.626, 1.201 |
| | Interest—7 | - 0.295 (0.181) | 0.744 | 0.522, 1.061 |
| | Interest—8 | - 0.561 (0.183) ** | 0.570 | 0.399, 0.816 |
| **Differentiation** | No Disclaimer—New Guideline | - 0.194 (0.142) | 0.824 | 0.623, 1.088 |
| $N_{ID}$ = 1109 | Disclaimer—New Guideline | - 0.195 (0.142) | 0.823 | 0.623, 1.087 |
| | Text order—Färber First | 0.516 (0.108) *** | 1.676 | 1.358, 2.071 |
| | Age | - 0.008 (0.004) * | 0.992 | 0.985, 0.999 |
| | Sex—Male | - 0.248 (0.108) * | 0.781 | 0.631, 0.965 |
| | Education—Medium | 0.005 (0.136) | 1.005 | 0.770, 1.312 |
| | Education—High | 0.192 (0.132) | 1.212 | 0.936, 1.571 |
| | Interest—5 | 0.125 (0.169) | 1.133 | 0.813, 1.578 |
| | Interest—6 | 0.089 (0.170) | 1.093 | 0.783, 1.527 |
| | Interest—7 | 0.104 (0.183) | 1.109 | 0.775, 1.587 |
| | Interest—8 | - 0.201 (0.182) | 0.818 | 0.572, 1.169 |
| **Causality of Effects** | Random Effect Variance (Participant) | 0.496 | - | - |
| $N_{obs}$ = 2241 | No Statement—New Guideline | - 0.181 (0.001)*** | 0.834 | 0.834, 0.835 |
| $N_{ID}$ = 1140 | Causality Statement—New Guideline | 0.100 (0.086) | 1.105 | 0.933, 1.309 |
| | Summary–Färber | - 0.125 (0.001) *** | 0.883 | 0.883, 0.883 |
| | Text Order–Färber First | 0.172 (0.079) * | 1.188 | 1.018, 1.387 |
| | Age | - 0.018 (0.001) *** | 0.982 | 0.982, 0.983 |
| | Sex–Male | 0.098 (0.079) | 1.104 | 0.945, 1.288 |
| | Education—Medium | 0.206 (0.090) * | 1.229 | 1.029, 1.467 |
| | Education—High | 0.690 (0.001) *** | 1.994 | 1.994, 1.995 |
| | Interest—5 | 0.085 (0.107) | 1.088 | 0.883, 1.343 |
| | Interest—6 | - 0.011 (0.105) | 0.989 | 0.806, 1.215 |
| | Interest—7 | - 0.162 (0.117) | 0.851 | 0.677, 1.070 |
| | Interest—8 | - 0.306 (0.001) *** | 0.737 | 0.736, 0.737 |

(*Continued*)

**Table 2.** (Continued)

| Outcome | Model Parameter | EST (SE) | OR | 95% CI |
|---|---|---|---|---|
| **Knowledge on CAMA** | No CAMA PLS—New Guideline | - 0.262 (0.167) | 0.769 | 0.554, 1.067 |
| $N_{ids}$ = 657 | CAMA PLS—New Guideline | 0.478 (0.167) ** | 1.613 | 1.163, 2.240 |
| | Text order—Färber First | - 0.111 (0.138) | 0.895 | 0.682, 1.173 |
| | Age | - 0.008 (0.139) | 0.992 | 0.983, 1.002 |
| | Sex—Male | 0.008 (0.139) | 1.008 | 0.768, 1.323 |
| | Education—Medium | 0.170 (0.176) | 1.186 | 0.840, 1.673 |
| | Education—High | 0.930 (0.177) *** | 2.534 | 1.792, 3.591 |
| | Interest—5 | - 0.147 (0.217) | 0.863 | 0.564, 1.321 |
| | Interest—6 | - 0.135 (0.217) | 0.874 | 0.571, 1.335 |
| | Interest—7 | - 0.137 (0.225) | 0.872 | 0.560, 1.356 |
| | Interest—8 | - 0.033 (0.237) | 0.968 | 0.607, 1.540 |
| **Funding** | Random Effect Variance (Participant) | 1.636 | - | - |
| $N_{obs}$ = 2712 | Version—New Guideline | - 0.013 (0.130) *** | 0.987 | 0.765, 1.274 |
| $N_{ID}$ = 1380 | Summary—Färber | 0.326 (0.072) | 1.386 | 1.203, 1.597 |
| | Text order—Färber First | 0.177 (0.010) *** | 1.193 | 0.981, 1.451 |
| | Age | - 0.015 (0.003) | 0.985 | 0.979, 0.992 |
| | Sex—Male | - 0.151 (0.101) *** | 0.860 | 0.705, 1.047 |
| | Education—Medium | 0.595 (0.127) *** | 1.814 | 1.415, 2.325 |
| | Education—High | 1.063 (0.127) | 2.894 | 2.254, 3.715 |
| | Interest—5 | 0.258 (0.157) | 1.295 | 0.951, 1.763 |
| | Interest—6 | 0.231 (0.158) | 1.260 | 0.925, 1.717 |
| | Interest—7 | 0.286 (0.170) | 1.331 | 0.954, 1.856 |
| | Interest—8 | - 0.122 (0.172) | 0.885 | 0.631, 1.241 |
| **Conflict of Interest** | Random Effect Variance (Participant) | 1.561 | - | - |
| $N_{obs}$ = 2675 | Version—New Guideline | - 0.170 (0.127) | 0.844 | 0.657, 1.083 |
| $N_{ID}$ = 1377 | Summary—Färber | 0.757 (0.073) *** | 2.132 | 1.848, 2.459 |
| | Text order—Färber First | 0.062 (0.098) | 1.064 | 0.879, 1,289 |
| | Age | - 0.005 (0.003) | 0.995 | 0.988, 1.001 |
| | Sex—Male | - 0.004 (0.099) | 0.953 | 0.785, 1.156 |
| | Education—Medium | 0.664 (0.125) *** | 1.943 | 1.522, 2.482 |
| | Education—High | 1.394 (0.126) *** | 4.032 | 3.146, 5.167 |
| | Interest—5 | 0.288 (0.155) | 1.334 | 0.984, 1.809 |
| | Interest—6 | 0.321 (0.155) * | 1.379 | 1.017, 1.869 |
| | Interest—7 | 0.117 (0.167) | 1.124 | 0.811, 1.558 |
| | Interest—8 | - 0.133 (0.169) | 0.875 | 0.628, 1.219 |

Note. $N_{obs}$ = Number of observations, $N_{ID}$ = Number of participants who provided ratings, EST = estimates for random effect variance of participants (causality of effects, funding, conflict of interest) and regression coefficients B for model parameters, SE = Standard Error, OR = Odds Ratio, 95% CI = 95% confidence interval for the odds ratio, "Relationship" = Knowledge on the relationship between meta-analysis and PLS, "Extent of Evaluation" = Knowledge on the extent of evaluation of research findings in the PLSs, "Differentiation" = Knowledge on the differentiation between PLS authors and meta-analysis authors, "Causality of Effects" = Knowledge on the causal interpretation of effects, "Knowledge on CAMA" = Knowledge on CAMA PLS, "Funding" = Knowledge on Funding, "Conflict of Interest" = Knowledge on Conflict of Interest, "Summary—Färber" = Knowledge when questions were asked after Färber and Rosendahl compared to Barth et al., "Text order—Färber first" = Knowledge when Färber and Rosendahl was presented as the first PLS

\* p < .05

\*\* p < .01

\*\*\* p < .001.

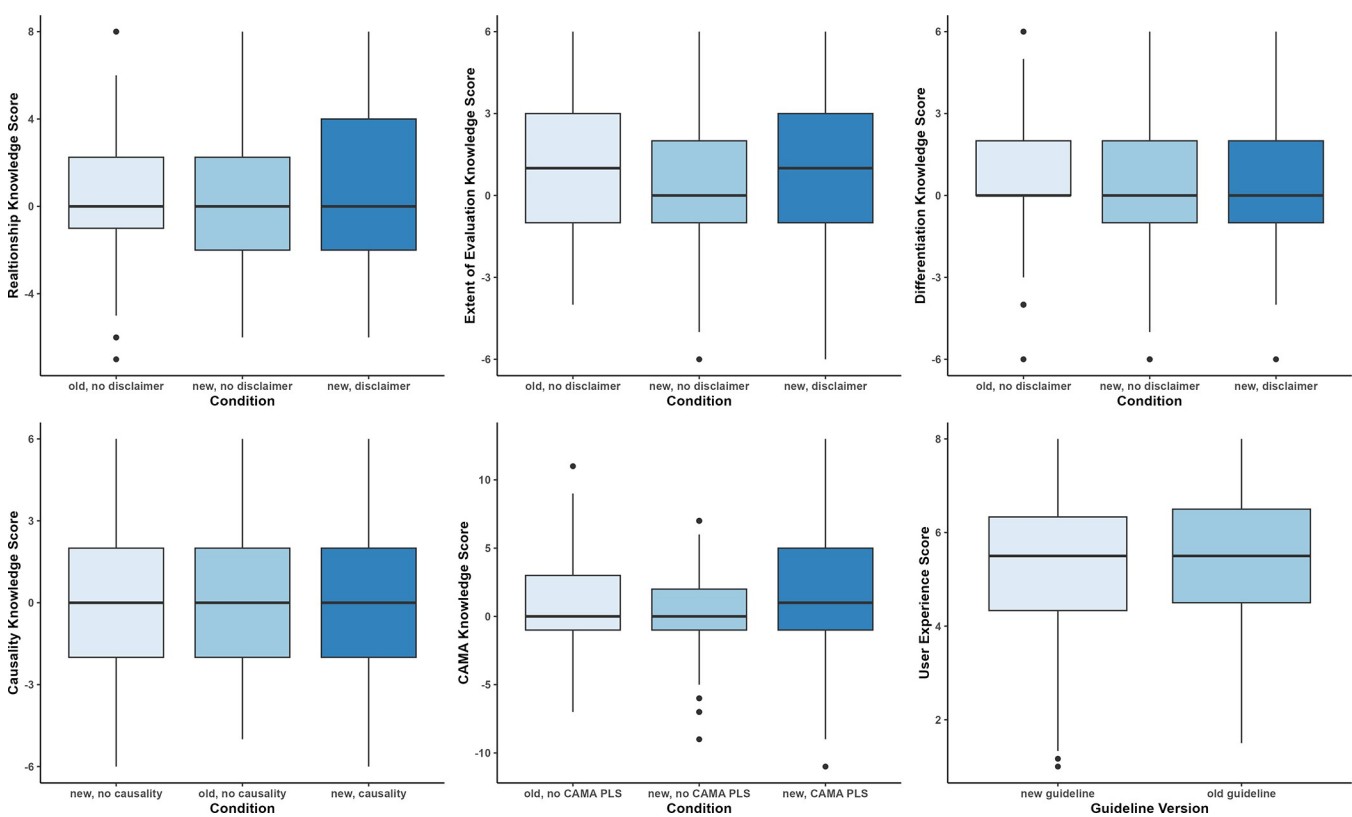

**Fig 1. Boxplots representing median knowledge item scores and upper/lower quartiles.** Boxplots represent the scores for H1 to H5 (top left to bottom middle) and overall user experience for H6 (bottom right). Categories on the x axis represent disclaimer conditions (H1-H3), causality statement conditions (H4), CAMA PLS conditions (H5) or guideline version (H6).

New Guideline" & "Disclaimer, New Guideline") as predictors. Due to a technical error, differentiation knowledge scores for the PLS based on resilience by Färber and Rosendahl could not be interpreted, hence research summary type had to be excluded as a control variable. To account for the influences of the text position of Barth et al. in the study, text order was instead introduced as a control variable. The overall model was significant compared to a null model, $\chi^2$ (11) = 45.03, $p <. 001$, with $R^2$ = .041. The guideline version or presence or absence of disclaimers had no significant impact on differentiation knowledge scores (No Disclaimer, New Guideline: OR = 0.82, 95%CI [0.62, 1.09], $p$ = .172; Disclaimer, New Guideline: OR = 0.82, 95%CI [0.62, 1.09], $p$ = .170). Furthermore, whether or not a PLS based on the new guideline included a disclaimer did not result in significant differentiation knowledge score differences ($Mdn_{\text{No Disclaimer}}$ = 0 vs. $Mdn_{\text{Disclaimer}}$ = 0, $W$ = 94509, $p$ = .978). Laypeople's knowledge on the differentiation between PLS and meta-analysis authors remained similar, both when the new guideline PLS with a disclaimer was compared to the old guideline PLS (H3a) and the regular new guideline PLS (H3b). H3a and H3b can thus be rejected.

## H4: Guideline version, causality statement and readers' knowledge on the causality of effects

H4 was analyzed via cumulative link mixed models. Readers' knowledge on the causality knowledge item was entered as a criterion, and due to its repeated measurement, a random intercept for individual readers was added to the model. Furthermore, two separate predictors

were entered to account for possible combinations of the guideline version and the use of a causality statement ("No Causality Statement, New Guideline" and "Causality Statement, New Guideline"). Also, we again included the above-mentioned control variables. The overall model fit was superior compared to a null model, $\chi^2$ (13) = 143.4, $p <.$ 001, with $R^2$ = .063. A presentation of PLS based on the new guideline without a causality statement significantly predicted a lower likelihood of correct answers among readers, OR = 0.83, 95%CI (0.83, 0.83), p < .001. There was no significant effect for PLSs based on the new guideline including a causality statement, OR = 1.11, 95%CI (0.93, 1.31), p = .246. Overall, participants showed poorer causality knowledge scores when solely new guideline PLSs were presented, and the effect was offset when a causality statement was included. In line with this, changing the reference level for the causality statement to "No Causality Statement, New Guideline" resulted in an overall significant model compared to the null model, $\chi^2$ (13) = 143.41, $p <.$ 001, $R^2$ = .063, in which readers were significantly more likely to correctly answer causality knowledge items when they received a PLS based on the new guideline including a causality statement, OR = 1.32, 95%CI (1.11, 1.58), p < .01. To summarize, presenting laypeople with a PLS based on the new guideline including a causality statement did not result in higher causality knowledge scores compared to the old guideline (H4a), but did improve causality knowledge compared to a PLS based on the regular new guideline (H4b). Thus, the results do not support H4a, but confirm H4b.

## H5: Guideline version, CAMA information and readers' knowledge on CAMA

Next, an ordinal logistic regression model was computed to examine H5. Readers' CAMA knowledge scores were entered into the model as criterion, and two variables specifying the guideline version and the use of additional CAMA elements ("No CAMA PLS, New Guideline", "CAMA PLS, New Guideline") served as predictors. Again, the aforementioned control variables and text order were included. Compared to a null model, the overall model offered a significantly better fit to participants' data, $\chi^2$ (11) = 54.09, $p < .001$, $R^2$ = .079. If readers received a PLS based on the new guideline version without CAMA elements, this did not influence their knowledge on CAMA, OR = 0.77, 95%CI (0.55, 1.07), p = .117. However, when they read through PLSs based on the new guideline including CAMA elements, this significantly increased their likelihood of selecting correct answers, OR = 1.61, 95%CI (1.16, 2.24), p < .01. Overall, participants were able to answer CAMA knowledge items more correctly if they were provided with PLSs based on the new guideline including a disclaimer. Additional Wilcoxon rank sum tests confirmed that readers' CAMA knowledge score was overall higher when PLSs based on the new guideline were provided ($Mdn_{\text{Old Guideline}}$ = 0 vs. $Mdn_{\text{New Guideline}}$ = 1, $W$ = 23613, $p < .05$). Additionally, their knowledge was higher when PLSs based on the new guideline included CAMA elements ($Mdn_{\text{No CAMA PLS}}$ = 0 vs. $Mdn_{\text{CAMA PLS}}$ = 1, $W$ = 17375, $p < .001$). Therefore, including additional information on CAMA in PLSs based on the new guideline was associated with higher laypeople knowledge on CAMA both in comparison to the old guideline (H5a) and the regular new guideline without CAMA information (H5b). The analyses thus lend support to H5a and H5b.

## H6: Guideline version and user experience

Participants' overall user experience (i.e., a mean score of their ratings of accessibility, understanding and empowerment) with the new guideline version was compared to the old guideline version in order to ensure that the revisions between versions did not result in a lower user experience. To that end, two-sided non-inferiority tests with an upper equivalence limit

of $\Delta = .20$ were carried out. For participants' overall user experience scores, the test reached significance, $\Delta = .20$, t(1339) = - 0.87, $p < .05$, suggesting that the new guideline is not inferior to the old guideline in terms of user experience. In line with H6, this confirms the a priori assumption that PLSs based on the new guideline would not lead to a significantly lower user experience compared to the old guideline among laypeople.

In addition to the confirmatory analyses reported thus far, multiple additional analyses were conducted to explore further research questions:

### RQ1: Guideline version and knowledge on funding

To examine whether the guideline versions used to create PLSs were associated with differences in readers' knowledge on funding, we once again employed cumulative link mixed models, with a random intercept for individual readers. Readers' knowledge on funding served as the criterion, while the guideline version ("New Guideline" vs. "Old Guideline") was specified as the predictor. As previously, research summary type, text order, participant's age, sex, educational background and interest ratings served as control variables. The overall model provided a superior fit to the data compared to a null model, $\chi^2$ (12) = 331.31, $p < .001$, with $R^2 = .117$. No significant influence of the guideline version on the likelihood to answer funding-questions was found, OR = 0.99, 95%CI (0.77, 1.27), p = .922, suggesting that the guideline version used to present PLSs had no influence on participants' knowledge on funding.

### RQ2: Guideline version and knowledge on conflict of interest

Whether readers' knowledge on COI was influenced by the guideline version was investigated via cumulative link modeling, once again with a random intercept for individual readers due to repeated measurements and the nested data structure. Knowledge on COI served as a criterion, while the guideline version ("New Guideline" vs. "Old Guideline") was entered into the model as a predictor. The same control variables as in RQ1 were also included in the model. The overall model fit was superior to a baseline null model, $\chi^2$ (12) = 437.10, $p < .001$, with $R^2 = .152$. The type of guideline version used for the creation of PLSs showed no significant impact on participants' overall knowledge on COI, OR = 0.84, 95%CI (0.66, 1.08), $p = .183$. There was thus no major difference in participants' COI knowledge scores between the new and the old guideline version.

### RQ3: Guideline version and readers' assessment of PLS author and meta-analysis author trustworthiness

As a final exploratory question, we examined the impact of the new and the old PLS writing guideline on participants' perceptions of PLS authors' and meta-analysis authors' trustworthiness via linear regression models. Three separate regression models were set up, with METI expertise, integrity and benevolence as respective criteria. Guideline version ("New Guideline" vs. "Old Guideline") and METI target ("Summary Authors" vs. "Study Authors") were entered into each model as predictors. Since the METI was presented after the second PLS, the research summary type of PLSs presented at the second position (i.e., "Färber and Rosendahl" vs. "Barth et al.") was entered as a control variable, together with participants' sex, age, educational attainment level and interest. The regression models reached significance for the METI outcomes of expertise, $F(8,1345) = 8.92$, $R^2 = .045$, $p < .001$, integrity, $F(8,1351) = 10.42$, $R^2 = .053$, $p < .001$, and benevolence $F(8,1347) = 11.59$, $R^2 = .059$, $p < .001$. No significant influences of the METI target on trustworthiness ratings were found, $\beta = .014 - .100$, all $ps \geq .108$. Similarly, the guideline version employed to create the PLSs did not significantly predict readers' trustworthiness ratings, $\beta = .032 - .092$, all $ps \geq .255$. In other words, whether readers

evaluated the trustworthiness of PLS authors or meta-analysis authors did not predict trust-worthiness differences. Furthermore, the old and new guideline versions were not associated with different trustworthiness scores.

## Discussion

### Summary of findings

The present study aimed to contrast PLSs of psychological meta-analyses based on two differ-ent writing guideline versions in terms of laypeoples' knowledge acquisition and user experi-ence. Our study therefore provides insights into the use of PLSs as a means to make research more accessible to laypeople and into the effects of specific textual elements such as a dis-claimer, a causality statement and information on CAMA on laypeoples' knowledge gain and reading experience.

Contrary to our expectations, the inclusion of a disclaimer on the extent of evaluation as well as a statement on the causality of effects often did not affect laypeoples' knowledge acqui-sition. This pattern could be observed both when comparing PLSs based on the new guideline including these elements with PLSs based on the old guideline (H1a, H2a, H3a, H4a) as well as the regular new guideline (H1b, H3b). A few explanations can be considered to interpret these findings: First, the basic PLSs created with the new guideline displayed a higher word count and were thus considerably longer compared to PLSs based on the old guideline (Barth et al.$_{new}$ = 982 vs. Barth et al.$_{old}$ = 863, Färber & Rosendahl$_{new}$ = 836 vs. Färber & Rosendahl$_{old}$ = 754), resulting in a length increase of about 17% for Barth et al. and 11% for Färber and Rosendahl. Including a disclaimer and causality statement further increased text length (Fär-ber & Rosendahl$_{new-complete}$ = 1,097, Färber & Rosendahl$_{new-complete}$ = 954), leading to length increases of about 31% for Barth et al. and 27% for Färber and Rosendahl. Drawing on cogni-tive load theory [34, 35], it could be argued that an increased text length associated with the new guideline may have introduced additional extraneous load for readers, which may have counteracted potential knowledge acquisition benefits, i.e. germane load, introduced by the disclaimer or causality statement. This may have introduced difficulties in knowledge reten-tion for readers and resulted in no major knowledge score differences for some of our hypoth-eses. Additionally, the disclaimer was presented at the end of the PLS. Readers' interest and attention to information may have decreased during reading, and the final passage interpreting the meta-analytic results may have indicated an end of relevant information to participants. Consequently, they may have simply skimmed the disclaimer without processing its content on a more elaborate level.

In addition, two findings are noteworthy: Compared to the regular new guideline, combin-ing a PLS based on the new guideline with a disclaimer resulted in improved knowledge on the extent of evaluation (H2b). A possible explanation for these results may be that the new guide-line immediately made the distinction between meta-analysis authors and PLS authors visible in the introduction. PLSs written according to the new guideline also more frequently used the term "Übersichtsarbeit" (review or review paper, a term we selected to refer to the original meta-analysis) in subheadings. This may have introduced additional information and ambigu-ity to laypeople when reading solely the regular new guideline, which may have only cleared up once the difference between PLS authors and meta-analysis authors was elaborated upon in the disclaimer statement. As a result, the PLSs based on the regular new guideline may have caused poorer knowledge gain compared to PLSs based on the new guideline with a disclaimer.

Furthermore, including a causality statement in the new guideline enabled participants to answer the knowledge item on the causality of effects more correctly (H4b). Related to these

results, it should be pointed out that the new guideline also contained a newly added section on limitations of the summarized meta-analysis. This may automatically make answers about the causal interpretation of effects and the interpretation of the reported findings more difficult for laypeople compared to PLSs based on the old guideline. It seems conceivable that the reduced performance in causality knowledge is only ameliorated once an additional causality statement informs laypeople about the inferences that can be drawn for the meta-analytic results. Thus, laypeoples' reduced performance on a causality knowledge item when reading the regular new guideline may, again, be associated with an increase in information and ambiguity.

Participants also demonstrated higher levels of knowledge regarding community augmented meta-analyses (CAMA) when a PLS contained additional information on CAMA. This pattern emerged both when a PLS based on the new guideline with CAMA elements was compared to the old guideline (H5a) and the regular new guideline (H5b). These results seem promising and suggest that including specific details about the methodology and process of CAMA in PLSs can enhance laypeople's understanding, even when it comes to more complex research findings. Generalizing one step further, providing such additional information may be viable in PLSs to reduce accessibility obstacles for non-experts and support them in grasping key concepts.

In terms of overall user experience, no significant differences were observed between the old and new versions of the PLS guideline (H6). Thus, suggestions for improvement provided by experts in the context of the first guideline evaluation, which partly increased PLS length, could be implemented without resulting in a poorer reading experience for laypeople. Both guideline versions can be considered effective in this regard.

Lastly, we did not find any effects of the two guideline versions on participants' knowledge on funding or COI as well as any differences in trustworthiness ratings between PLS authors and meta-analysis authors. It should be noted that both guideline versions contained almost identical information on these issues, but at different positions. Our results suggest that the exact position of these information in the text likely had no influence. As for similar trustworthiness ratings, previous research has been able to demonstrate that readers frequently disregard source information when reading through single texts (c.f. [36]). It seems plausible that laypeople may have primarily focused on the text content in the present study, rather than the information outlining the differentiation between PLS authors and meta-analysis authors.

## Demographic effects

Generally speaking, two consistent effects of demographic variables on item scores could be observed in the present study. First, a higher reader age was associated with lower scores on knowledge items on the relationship between the PLS and the corresponding meta-analysis (H1), the extent of evaluation (H2), the differentiation between PLS authors and meta-analysis authors (H3) and the causality of effects (H4, see Table 2). Reasons for this may include a poorer memory performance associated with higher age (e.g. in item recognition tasks, see [37]), or a potential greater familiarity of younger laypeople with short research summaries or info texts prevalent in online contexts. And second, a higher educational attainment level (Mittlere Reife or Abitur) generally predicted higher scores regarding knowledge on the relationship between the PLS and the corresponding meta-analysis (H1), on the extent of evaluation (H2), on the causality of effects (H4) and on CAMA (H5, see Table 2). Given that both individual student characteristics such as socio-economic status (SES) and science self-efficacy as well as school factors such as school-level SES and parental involvement can affect scientific literacy [38], and that these variables likely reach higher values in connection to secondary school

types such as "Realschule" or "Gymnasium/Gesamtschule" in Germany, a higher level of scientific literacy may be the driving force behind these findings.

Based on these observations, it may be worthwhile to ask the following question: How can one support older laypeople or laypeople with a comparatively lower level of education in processing and understanding psychological PLSs? After all, both these characteristics seem to be predictive of lower scores on multiple knowledge items. One approach could be to not only include textual information in PLSs, but to also rely on elements such as infographics, or to develop PLS formats including both orally narrated and visually presented information. Based on the cognitive theory of multimedia learning [39], this approach is likely to have benefits for both laypeople's encoding and retrieval of information, e.g. by reducing the information load that requires processing. The present study was first and foremost focused on textual elements. in order to create an empirically validated guideline without focusing on needs of different user groups. As such, we did not take multimedia learning into account. Future research would benefit from drawing on multimedia principles to make PLSs more effective and to address particular needs of specific demographic groups.

## Implications for future research

The current study offers multiple considerations for the field of science communication and future research on PLSs in psychology. As mentioned before, the KLARpsy-guideline compiled existing evidence and experimentally tested variations in PLS formats to provide an empirically validated blueprint for creating PLSs of psychological meta-analyses. The overarching aim is to create PLSs that facilitate laypeople the access to complex scientific publications and to support them in grasping the core concepts of research works. Two implications therefore seem noteworthy: First, laypeople seem to be able to grasp key concepts of a PLS even when further complexity is introduced via multiple authorship levels (i.e., PLS authors and meta-analysis authors), and to understand hints about the fact that a PLS was not evaluated in terms of methodical rigor or scientific correctness by the PLS authors. This implies that even brief science communication formats can offer laypeople this information without negatively affecting knowledge gain or user experience. And second, it seems possible to use PLSs to introduce relatively complex methodologies such as CAMA to laypeople. When introduced via disclaimers and additional explanations in PLSs, this may increase readers' knowledge gain compared to PLSs that omit such concepts or only mention them briefly. We would thus like to encourage future research on PLS to further investigate these textual elements.

Another point worth investigating may be the role that information on research limitations in PLSs plays for laypeople's knowledge gain. Introducing such information via a passage in the regular new guideline may have caused more ambivalence and uncertainty for laypeople. The result may have been a poorer performance regarding the causal interpretation of effects. Future PLS research could consider this question, e.g. by comparing text conditions with different types of causality statements against one another. This has the potential to further outline the effects of limitation statements on laypeople and how these limitation statements can be buffered.

Further PLS development would also benefit from a more in-depth analysis of user experience and user behavior. For example, employing eye-tracking technology [40] could help to identify laypeople's gaze patterns, thereby improving understanding about which PLS passages laypeople focus the most attention on and which passages they may merely skim. Similarly, recognition and recall memory tests could be used to further explore differences between future PLS versions. This could even be combined with time-delayed follow-up tests (e.g. after 2, 4 or 6 weeks) to explore how well laypeople retain information included in PLS in their

long-term memory. And, finally, examining how practitioners such as clinical psychologists, counselors, child development specialists or recruiters as well as policymakers draw on PLSs and which PLS aspects may prove beneficial when transferring information into practical work remains a worthwhile question. Although initial research on this issue exists for systematic reviews [41, 42] and points towards the benefits of said reviews for practitioners, further studies and more thorough examinations have the potential to uncover how PLSs can be turned into a more effective support-tool for decision-making. Investigating practitioners' or policymakers' reactions to PLSs, e.g. in the context of a decision-making task, may help to illuminate factors that facilitate or impede PLS use among these groups. All of these approaches have the potential to further uncover how laypeople interact with PLS and which text features may be of particular importance for knowledge gain and informed decision-making.

## Strengths and limitations

There are three major strengths of our study we would like to point out:

First, our study was preregistered, which enhances the transparency of our research and increases the validity of our findings. Second, we collected data from a large sample of the German-speaking general population, which increases the generalizability of our findings beyond a purely academic context. To achieve this, we aimed for a balanced sample, with an approximately equal number of participants in each demographic category. Third, we employed standardized PLSs derived from published and peer-reviewed psychological research. This enhances the ecological validity of our study and ensures that our findings may be applied to practical contexts where PLSs of psychological research are provided to laypeople.

However, one main limitation is that we were not able to control for variables apart from those targeted via our experimental variations and covariates. Due to the evaluation process during guideline development and for reasons of practicality during the PLS writing process, it was not possible to maintain constant control over all other influencing factors (e.g., text length). This limitation may have introduced additional variability into the study results, making it challenging to isolate the effects of our specific manipulations.

Moreover, a technical error which occurred during the study has to be taken into account with respect to the interpretation of knowledge on the differentiation between PLS authors and meta-analysis authors (H3). The analysis of the respective knowledge item was only possible for the PLS based on Barth et al., and specific text effects for Färber and Rosendahl cannot be ruled out. Finally, we employed newly created knowledge items specifically designed for the assessment of knowledge in the context of PLSs in our study. As no validated short knowledge items for PLSs are currently available, this represents a crucial limitation. Future research could aim to develop and validate a scale specifically tailored for assessing knowledge gain after reading PLSs or short research summary formats.

## Conclusions

In line with earlier research, our study corroborates the potential of PLSs of psychological meta-analyses to bridge the gap between academic research and lay audiences. By systematically testing annotations and suggestions from an expert user survey and individual user feedback in a large target group sample, we complied to a fundamental requirement for evidence-based, accessible and high-quality PLSs. The inclusion of additional information on complex methodologies, such as CAMA, can enhance laypeoples' knowledge acquisition regarding more complex and innovative scientific methods. However, the effectiveness of other elements, such as information on authorship of PLS and meta-analysis (a disclaimer) and a causality statement, may require further investigation. These findings contribute to the ongoing efforts

to improve science communication and to make psychological research more accessible and comprehensible to the general public.

## Supporting information

**S1 File. Original German PLSs and English translations.** https://doi.org/10.23668/psycharchives.14217.
(PDF)

**S2 File. Dataset.** https://doi.org/10.23668/psycharchives.14218.
(CSV)

**S3 File. Codebook.** https://doi.org/10.23668/psycharchives.14218.
(CSV)

**S4 File. Complete R Markdown.** https://doi.org/10.23668/psycharchives.14219.
(ZIP)

**S5 File. Shortened R Markdown.** https://doi.org/10.23668/psycharchives.14219.
(ZIP)

## Acknowledgments

We would like to acknowledge Lea Stulz for her support in proofreading and preparing the final version of the manuscript.

## Author Contributions

**Conceptualization:** Mark Jonas, Martin Kerwer, Marlene Stoll, Gesa Benz, Anita Chasiotis.

**Data curation:** Mark Jonas, Martin Kerwer.

**Formal analysis:** Mark Jonas, Martin Kerwer.

**Investigation:** Mark Jonas, Martin Kerwer, Marlene Stoll, Anita Chasiotis.

**Methodology:** Mark Jonas, Martin Kerwer, Marlene Stoll, Gesa Benz, Anita Chasiotis.

**Project administration:** Anita Chasiotis.

**Resources:** Anita Chasiotis.

**Software:** Mark Jonas, Martin Kerwer.

**Supervision:** Anita Chasiotis.

**Validation:** Mark Jonas, Martin Kerwer, Marlene Stoll, Gesa Benz, Anita Chasiotis.

**Visualization:** Mark Jonas, Martin Kerwer.

**Writing – original draft:** Mark Jonas, Martin Kerwer, Marlene Stoll, Gesa Benz, Anita Chasiotis.

**Writing – review & editing:** Mark Jonas, Martin Kerwer, Marlene Stoll, Gesa Benz, Anita Chasiotis.

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
