## [Decision Letter · Decision Letter 0]

19 Dec 2023

PONE-D-23-30389Equivalent User Experience and Improved Community Augmented Meta-Analyses Knowledge for a New Version of a Plain Language Summary GuidelinePLOS ONE

Dear Dr. Jonas,

First let me start by apologizing for how long it took to get you a decision. Potential reviewers seem pinched for time and overwhelmed with their current workload. I am thankful that two reviewers agreed and made thoughtful reviews of your paper. As you can see, they are both extremely positive about your manuscript. Your dedication to open science was noted and is much appreciated. The reviewers have suggested a few changes that should be easy to make, and some suggestions to think about. I think you should be able to address these concerns easily on a resubmission. I would add one point to their reviews. In the end of each Results section, you mention if a numbered hypothesis is supported. Instead of just listing the number, a short textual reminder of what that hypothesis is would be helpful for the reader. If there are any changes you are not willing to make, please just provide a justification or alternative plan. I am happy to be flexible on your resubmission deadline given how long it took for me to get you a decision.So in official words, we invite you to submit a revised version of the manuscript that addresses the points raised during the review process. Please submit your revised manuscript by Feb 02 2024 11:59PM. If you will need more time than this to complete your revisions, please reply to this message or contact the journal office at plosone@plos.org. Please include the following items when submitting your revised manuscript:A rebuttal letter that responds to each point raised by the academic editor and reviewer(s). You should upload this letter as a separate file labeled 'Response to Reviewers'.A marked-up copy of your manuscript that highlights changes made to the original version. You should upload this as a separate file labeled 'Revised Manuscript with Track Changes'.An unmarked version of your revised paper without tracked changes. You should upload this as a separate file labeled 'Manuscript'.If applicable, we recommend that you deposit your laboratory protocols in protocols.io to enhance the reproducibility of your results. Protocols.io assigns your protocol its own identifier (DOI) so that it can be cited independently in the future. For instructions see: https://journals.plos.org/plosone/s/submission-guidelines#loc-laboratory-protocols. Additionally, PLOS ONE offers an option for publishing peer-reviewed Lab Protocol articles, which describe protocols hosted on protocols.io. Read more information on sharing protocols at https://plos.org/protocols?utm_medium=editorial-email&utm_source=authorletters&utm_campaign=protocols.

We look forward to receiving your revised manuscript.

Kind regards,

Jessecae Marsh, PhD

Academic Editor

PLOS ONE

“The study was funded by internal ZPID funds. The authors received no specific funding for this work.”

Reviewers' comments:

Reviewer's Responses to Questions

**Comments to the Author**

1. Is the manuscript technically sound, and do the data support the conclusions?

Reviewer #1: Partly

Reviewer #2: Yes

2. Has the statistical analysis been performed appropriately and rigorously? 

Reviewer #1: Yes

Reviewer #2: Yes

3. Have the authors made all data underlying the findings in their manuscript fully available?

Reviewer #1: Yes

Reviewer #2: Yes

4. Is the manuscript presented in an intelligible fashion and written in standard English?

Reviewer #1: Yes

Reviewer #2: Yes

5. Review Comments to the Author

Reviewer #1: This paper is generally scientifically sound but with two notable issues that require clarification. I also have a handful of comments about things that would be helpful to clarify or highlight to aid the reader in interpreting and perhaps applying these results.

The first issue has to do with the nature of some of the comparisons in relation to the six conditions the authors used. In short, almost all of the significant results seem to be found in highly asymmetric comparisons, in which one group is 3x-5x the size of the group it's being compared to. For example, for H1a-H5a, all of the comparisons against the old guidelines are combinations of three conditions versus the one condition that saw the old guidelines. The CAMA analysis is the same thing in reverse: only one condition saw the CAMA statement, and it is compared against the five that did not. The authors should make clear whether this imbalance affects the interpretation of their results, whether it is incorporated into the underlying assumptions of their analytical approach, and perhaps whether these comparisons hold up when individual conditions are compared pairwise in post-hoc tests.

The second issue can be resolved by simply reporting a brief analysis or some additional descriptive information: were the 33% of participants who failed the awareness check evenly distributed across demographic groups and conditions? With regard to conditions, while the individual conditions vary in size by up to ~45 participants depending on the measure, there are no obvious patterns based on the specific factors being manipulated. The authors should present descriptive data or an analysis to demonstrate whether the same is true for the demographic factors considered, and if not, consider whether this impacts the interpretation of some of the effects of demographics (which were consistent throughout).

Those are my only scientifically substantive comments, but both should be relatively straightforward to address.

Regarding the presentation of this work, there are a number of details that I think the authors should clarify to help the reader understand the rationale and importance of this study.

First, was the goal to get a demographically representative sample, or a demographically balanced sample? Frankly I don't know the demographics of the German populace to say whether those are going to be the same or not, but based on the breakdown described by the authors, it sounds like they went for a balanced sample where each demographic group had roughly the same number of participants. However, they cite the fact that they use a representative sample as one of the strengths of this work on line 675. Is that truly representative of the proportions of these demographic groups in the German population as a whole? This is worth considering because it affects how this kind of work may be used in the future. There is an effect of age on almost every measure, for example, but if one age group is overrepresented in the general population, and it's the age group that showed overall worse performance, then perhaps this whole approach needs to be reconsidered.

Second, it was not clear why some measures or factors were important. For example, H1ab and H3ab could use some additional justification. Why is it important that lay readers understand the difference between the PLS and the meta-review? Why does it matter that they have different authors? What are the ramifications of these differences for the public's understanding of the underlying science? I have similar concerns about H5ab, and the entire CAMA question. It would be helpful if the authors explained why it is important to know whether the public is sensitive to these features, and what ramifications this sensitivity might have.

Finally, I think more should be said about the demographic effects, since these were highly consistent. Do you have any explanation for them? And, as discussed above, do they have ramifications for the usefulness of these guidelines for effective science communication with different populations?

Overall this is generally sound work and an interesting approach to science communication, but a few clarifications are needed.

Reviewer #2: Dear Authors,

I find your project to be very interesting and believe that your innovative approach to scientific communication has the potential to yield insights while paving the way for fresh avenues of research and refined methods of scientific dissemination. Allow me to share a few comments and suggestions to enhance the impact of your work. I hope you will find them useful.

(1) It appears that your project has invested considerable effort and resources into a complex research, yet it provides limited information regarding the practical applicability of PLS/the new guidelines. In your considerations for future research, it may be advantageous to incorporate methods that scrutinize the effectiveness of (the new) PLS using modern user experience (UX) tools such as eye tracking and memory tests. By streamlining your project, focusing on fewer hypotheses and research questions, and employing precise instruments, you could potentially uncover more influential and practical insights.

(2) Exploring the combination of textual and visual elements as a means to convey the essence of meta-analyses might prove more valuable, even if it deviates (slightly) from the new PLS standards. As such, it could limit the information load.

(3) The challenge in communicating the results of meta-analyses lies in their abstract nature, demanding a high degree of abstract thinking and prior knowledge. It would be advisable to consider the characteristics of your target audience in future research research. Understanding how meta-analytical findings impact professionals and policymakers could be pivotal, and including this in your discussion could enhance the (practical) relevance of your work.

(4) I appreciate your commitment to open science principles, even beyond the journal's requirements. However, the extensive code within your analysis script can be overwhelming for reviewers who have limited time to verify and reproduce your analyses. To address this, I recommend sharing code that facilitates a step-by-step reproduction of the analyses as presented in your manuscript, ensuring ease of identification and inspection of the most critical analyses.

I wish you the best of luck with your forthcoming projects!

6. PLOS authors have the option to publish the peer review history of their article (what does this mean?). If published, this will include your full peer review and any attached files.

Reviewer #1: **Yes: **Jonathan F. Kominsky

Reviewer #2: **Yes: **Jacek Buczny

---

## [Author Response · Author response to Decision Letter 0]

14 Feb 2024

We would like to thank the editorial manager and both reviewers for their time and effort thus far. As requested, a response letter has been included that adresses the points raised by the editorial manager and the reviewers.

---

## [Decision Letter · Decision Letter 1]

4 Mar 2024

Equivalent user experience and improved community augmented meta-analyses knowledge for a new version of a Plain Language Summary guideline

PONE-D-23-30389R1

Dear Dr. Jonas,

Thank you for so carefully addressing the reviewer concerns! We’re pleased to inform you that your manuscript has been judged scientifically suitable for publication and will be formally accepted for publication once it meets all outstanding technical requirements. 

Kind regards,

Jessecae Marsh, PhD

Academic Editor

PLOS ONE

Additional Editor Comments (optional):

Reviewers' comments:

Reviewer's Responses to Questions

**Comments to the Author**

1. If the authors have adequately addressed your comments raised in a previous round of review and you feel that this manuscript is now acceptable for publication, you may indicate that here to bypass the “Comments to the Author” section, enter your conflict of interest statement in the “Confidential to Editor” section, and submit your "Accept" recommendation.

Reviewer #1: All comments have been addressed

Reviewer #2: All comments have been addressed

2. Is the manuscript technically sound, and do the data support the conclusions?

Reviewer #1: Yes

Reviewer #2: Yes

3. Has the statistical analysis been performed appropriately and rigorously? 

Reviewer #1: Yes

Reviewer #2: Yes

4. Have the authors made all data underlying the findings in their manuscript fully available?

Reviewer #1: Yes

Reviewer #2: Yes

5. Is the manuscript presented in an intelligible fashion and written in standard English?

Reviewer #1: Yes

Reviewer #2: Yes

6. Review Comments to the Author

Reviewer #1: (No Response)

Reviewer #2: Dear Author(s),

Thank you for addressing all my recommendations. The new .Rmd file is transparent and easy to use. I do not have any further comments.

Good luck with your future projects!

7. PLOS authors have the option to publish the peer review history of their article (what does this mean?). If published, this will include your full peer review and any attached files.

Reviewer #1: **Yes: **Jonathan F. Kominsky

Reviewer #2: **Yes: **Jacek Buczny

---

## [Editor Report · Acceptance letter]

26 Apr 2024

PONE-D-23-30389R1 

PLOS ONE

Dear Dr. Jonas, 

I'm pleased to inform you that your manuscript has been deemed suitable for publication in PLOS ONE. Congratulations! Your manuscript is now being handed over to our production team.

Kind regards, 

on behalf of

Dr. Jessecae Marsh 

Academic Editor

PLOS ONE